# Quantification and Comparison of Nutritional Components in Oni Walnut (*Juglans ailanthifolia* Carr.), Hime Walnut (*Juglans subcordiformis* Dode.), and Cultivars

**Ritsuko Fukasawa** [1,2,3], **Taiki Miyazawa** [2,*], **Chizumi Abe** [2], **Maharshi Bhaswant** [2] **and Masako Toda** [1,2]

1   Graduate School of Agricultural Science, Tohoku University, Sendai 980-8555, Japan;
    fukasawa.ritsuko.p1@dc.tohoku.ac.jp (R.F.); masako.toda.a7@tohoku.ac.jp (M.T.)
2   New Industry Creation Hatchery Center (NICHe), Tohoku University, Sendai 980-8579, Japan;
    chizumi.abe.e5@tohoku.ac.jp (C.A.); cmaharshi@gmail.com (M.B.)
3   Department of Health and Nutrition, Tohoku Seikatsu Bunka University, Sendai 981-8585, Japan
*   Correspondence: taiki.miyazawa.b3@tohoku.ac.jp; Tel.: +81-22-795-3205

**Abstract:** Walnuts are rich sources of lipids and polyunsaturated fatty acids and are expected to promote health. There are two Japanese native walnut species: Oni walnut (*Juglans ailanthifolia* Carr.) and Hime walnut (*Juglans subcordiformis* Dode.). However, despite the fact that these Japanese native walnuts have long been consumed in local cuisine, their nutritional composition is largely unknown. This study aimed to assess the concentrations of total lipids, and fatty acid composition including polyunsaturated fatty acids, in the kernels of Oni walnut and Hime walnut. In addition, we assessed various aspects related to their nutritional and functional values, by measuring the total protein, amino acids, minerals, and total polyphenols. The concentrations of the measured compounds in the two native species were compared with those in the English walnut (*Juglans regia* L.), a globally recognized cultivar, and its counterpart cultivated in Japan, Shinano walnut (*Juglans regia* L.). The results showed that Oni walnut and Hime walnut contained significantly higher protein and minerals and lower lipid content than conventional cultivars. However, both Oni and Hime walnuts contained higher ratios of unsaturated fatty acids in total fatty acids. This study offers novel insights into the nutritional components of Oni and Hime walnuts, contributing to a deeper understanding of their nutritional value and potential applications as unique native walnut species. The findings of this study highlight the relationship between the different types of walnut species and their nutritional composition, and the value of native walnuts used in local cuisine, and will lead to new developments in functional foods from walnut species consumed around the world. It will contribute to the development of functional and processed foods by increasing the production of native walnut species, which are rich in protein, unsaturated fatty acids and minerals and by using them in local cuisines and health-promoting foods.

**Keywords:** amino acids; dietary fats; English walnut; fatty acids; Hime walnut; local cuisine; minerals; Oni walnut; polyphenols; Shinano walnut





## 1. Introduction

Walnut trees are known for their ability to grow in a broad climate range. Their adaptability to diverse climatic and soil conditions made them a staple cultivation since ancient times [1]. Walnut kernels are considered a valuable and nutritious resource for humans. Therefore, they have been incorporated into culinary cultures worldwide. Numerous cohort studies have previously investigated the health implications of walnut consumption [2]. Among these studies, particular attention has been directed towards understanding the interplay between the fatty acid composition and phytochemical content of walnuts, as well as their potentials to prevent and treat various diseases, including chronic inflammatory diseases, cardiovascular diseases, diabetes, and cancer [3,4]. To elucidate the health benefits

of walnuts, assessing the concentrations of nutritional components in each walnut type is crucial [5,6]. Current research on walnuts has focused on the function of a cultivated species known as English walnut (*Juglans regia* L.). Consequently, knowledge of the nutritional components, functional properties, and optimal cultivation techniques of this species has accumulated [7,8].

English walnut is the most widely cultivated walnut species worldwide [9]. Meanwhile, native walnut species have gained attention because of their natural abundance of phytochemicals, which enhance their resistance to environmental variations compared to cultivars. Native walnut species also exhibit resistance to external stressors, pests, and climate fluctuations [10]. Cultivation endeavors are gaining attention across various regions of Western Europe, notably as part of the BaSeFood program [11]. This research initiative, supported by the European Commission, is dedicated to sustainably harnessing the bioactive components present in traditional foods from the Black Sea region. Among the spectrum of native walnut species, the Black walnut (*Juglans nigra* L.) is one of the most widely consumed nuts in the United States [12]. Compared to English walnut, the Black walnut contains elevated concentrations of phenolic compounds (measured by high-performance liquid chromatography coupled with tandem mass spectrometry), amino acids (measured by high-performance liquid chromatography (HPLC)), iron (measured by inductively coupled plasma optical emission spectroscopy (ICP-OES)), and zinc (measured by ICP-OES) [5,13]. Furthermore, their fatty acid composition includes a higher proportion of unsaturated fatty acids (UFA). These distinctive attributes have propelled the Black walnut into the spotlight as a potential functional food. It has been used in sweets, bakery products, ice cream, cereals, and as a cooking ingredient in snacks due to its beneficial nutritional composition [8,12].

In Japan, Oni walnut (*Juglans ailanthifolia* Carr.) and Hime walnut (*Juglans subcordiformis* Dode.) are native walnut species that have adapted to the unique climate and are used as ingredients in local cuisine [14–16]. Oni walnut and Hime walnut have a rich flavor with little astringency. They grow mainly in eastern Japan and are used in various local dishes and sweets. Oni walnut is known from its hard shell, which has deep ridges on its surface. The name of 'Oni walnut' is named for its hard shell with deep ridges on its surface, reminiscent of Oni (a traditional Japanese monster) [17]. Hime walnut is considered as variant of the Oni walnut, with a thin, heart-shaped shell that is easy to break to extract the kernel [17]. However, despite growing curiosity regarding the potential health benefits of Oni walnut and Hime walnut, limited information on their nutritional composition is available [16]. Even fundamental aspects that are well-documented for other walnut species, such as the fatty acid composition and polyphenol content, remain largely unexplored in these native varieties [16].

In light of this context, the principal objective of this study was to elucidate the nutritional composition of Oni walnut and Hime walnut. To achieve this objective, we aimed to determine the crucial components, including the total lipid content, fatty acid composition, total protein, amino acid composition, total polyphenols, and mineral content. This comprehensive analysis provides crucial insights into the nutritional potential and prospective health benefits of Japanese native walnuts. The protection and expansion of native walnut vegetation will provide the opportunity to revitalize local industries through the development of functional and processed foods and will become a way to pass on local cuisine and food culture into the future.

## 2. Materials and Methods

First, we evaluated four walnut species (Table 1 and Figure 1), English walnut kernels were harvested in California, United States, in 2022 and were acquired from a street vendor in Tokyo, Japan. Shinano walnut is an English walnut cultivated in Japan, and was harvested in Nagano prefecture in September 2022. This variety was purchased in-shell from a direct sales outlet. Oni walnut and Hime walnut Japanese native walnut species were harvested from Yamagata prefecture in September 2022. These species were purchased

in-shell from the producer sales outlet. To ensure the integrity of the walnut nutritional components, all purchased walnuts were stored at −20 °C. In-shell walnuts were shelled the day before commencing the experiment, and their kernels were accurately weighed. Subsequently, these kernels were placed in sealed plastic bags and stored at −20 °C until immediately before the extraction process. This study was conducted during a storage period of two months, and the storage conditions and period of time guaranteed that the nutritional composition of walnut kernels was not affected [18,19]. The data for total lipid were calculated from five measurements, with triplicate measurements excluding the first and last. Total protein and amino acids were measured by duplicate. All other quantities were measured once.

**Table 1.** Walnut species measured in this study.

| Species | Botanical Name | Country | State or Prefecture | Color of Pellicle |
|---|---|---|---|---|
| Cultivar | English walnut | *Juglans regia* L. | United States | California (CA) | Extra Light |
| | Shinano walnut | *Juglans regia* L. | Japan | Nagano | Extra Light |
| Japanese native walnut | Oni walnut | *Juglans ailanthifolia* Carr. | Japan | Yamagata | Light Amber |
| | Hime walnut | *Juglans subcordiformis* Dode. | Japan | Yamagata | Light |

The color of the pellicle was decided in accordance with the regulations of the United States Department of Agriculture (USDA).

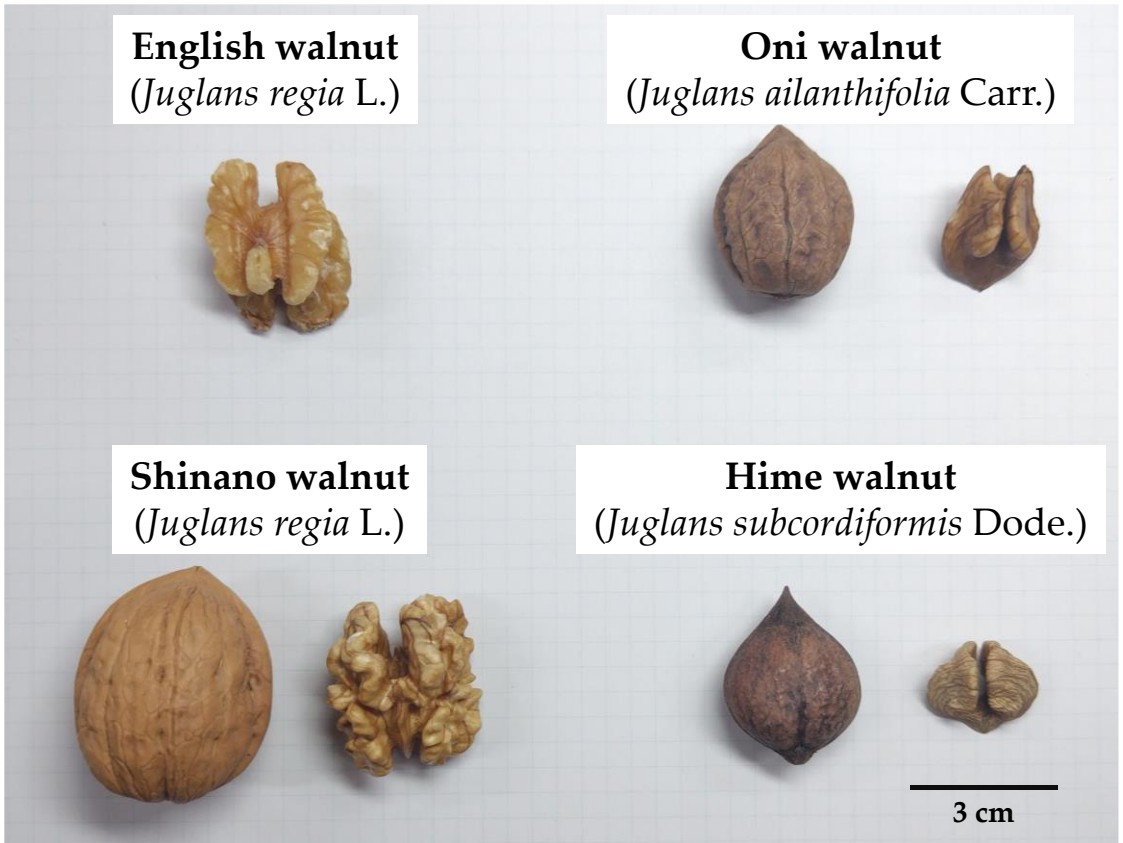

**Figure 1.** Photographs of walnut species measured in this study.

For subsequent analyses of the walnut nutritional components, all reagents employed were of HPLC grade or higher, unless otherwise specified. These reagents were purchased from the Fujifilm Wako Pure Chemical Corp. (Osaka, Japan) and Sigma-Aldrich Co., LLC. (St. Louis, MO, USA).

### 2.1. Measurement of the Total Lipid Content

To determine the total lipid contents, lipids in walnuts were extracted according to the method described by previous reports [20,21]. Briefly, walnut kernels were ground using a crush mixer (IFM-C20G, Iwatani Corp., Tokyo, Japan). The pulverized walnuts were further ground into super fine powder using a mortar and pestle. Fine powdered walnuts (2.5 g) were mixed with 7.5 mL of a 0.9% sodium chloride (NaCl) solution containing 1.0 mM ethylenediaminetetraacetic acid (EDTA). To this mixture, 30 mL of chloroform: methanol (2:1 *v/v*) was added. The mixture was shaken at 1500 rpm for 10 min at room temperature. The entire extraction process was conducted under a nitrogen atmosphere to maintain sample integrity. Subsequently, the mixture was subjected to centrifugation at 4 °C for 10 min at $850 \times g$, resulting in the separation of the components of the mixture, with the lower layer containing the extracted lipids. The collected lower layer was filtered through anhydrous sodium sulfate and dried using a rotary evaporator. To determine the total lipid content, the dried samples were allowed to stand for one hour in a vacuum desiccator. The sample weights were then precisely measured. All procedures were performed in a light-shielded environment, with round-bottom flasks containing the extracted lipids covered with aluminum foil for further analysis.

### 2.2. Evaluation of the Fatty Acid Composition

Following the measurement of total lipid content, the extracted lipids were subjected to fatty acid methylation. This transformation was performed using trifluoroborane methanol, following the previously reported method [22,23]. Subsequently, the methylated fatty acids were subjected to gas chromatography coupled with flame ionization detection (GC-FID, SIMADZU GC-2010, Shimadzu Corp., Kyoto, Japan) to analyze the fatty acid composition of the walnuts. The analytical conditions for GC-FID were set as follows: a sample injection volume of 1 μL, utilization of helium as the carrier gas, a column flow rate of 1.16 mL/min, a vaporization chamber temperature of 240 °C, and a split ratio of 10:1. A Phenomenex ZB-FAME column (Phenomenex Inc., Torrance, CA, USA) with internal diameter of 0.25 mm, length of 60.0 m, and film thickness 0.20 μm was used for the chromatographic separation. The oven temperature was programmed to incrementally rise from 120 °C to 260 °C, and the total analysis time was set at 57.17 min.

### 2.3. Measurement of the Total Protein Content

The quantification of total protein content in walnuts was conducted through combustion by employing the modified Dumas method [24,25]. Specifically, 0.3–0.4 g of vacuum freeze-dried and ground walnut kernels was accurately weighed and placed into a 2 mL quartz boat (Sumika Chemical Analysis Service, Ltd., Tokyo, Japan). Subsequently, samples were analyzed using a total nitrogen analyzer (SUMIGRAPH NC-TRINITY, Sumika Chemical Analysis Service, Ltd., Tokyo, Japan). Analytical conditions for the total nitrogen analyzer were set as follows: the reaction furnace temperature was maintained above 870 °C, the reduction furnace temperature was 600 °C, the detector temperature was 100 °C, the column temperature was 70 °C, the oxygen flow rate was $0.20 \pm 0.05$ L/min, and the helium flow rate was $80 \pm 5$ mL/min. A calibration curve was constructed using EDTA as the standard. The calculated amount of nitrogen in the samples was converted into protein content.

### 2.4. Measurement of the Amino Acid Content

Amino acids were quantified following the protocols as described previously by Wu et al. [26]. and Liu et al. [27]. To analyze arginine, lysine, histidine, phenylalanine, tyrosine, leucine, isoleucine, valine, alanine, glycine, proline, glutamic acid, serine, threonine, and aspartic acid, 0.5 g of walnut kernels was added to 20 mL of 20% hydrochloric acid containing 0.04% 2-mercaptoethanol. The mixture was defatted, sealed in a tube, and hydrolyzed at 110 °C for 24 h. Subsequently, the protein hydrolysate was brought up to a final volume of 100 mL with deionized water, and a 2 mL aliquot was concentrated

under reduced pressure until dry. Subsequently, 20 mL of sodium citrate buffer (pH of 2.2) was added to the dried sample to prepare the test solution, which was analyzed using a high-speed amino acid analyzer (LA8080, Hitachi High-Tech Science Corp., Tokyo, Japan). The analytical conditions for the high-speed amino acid analyzer were set as follows: the column consisted of Hitachi custom ion-exchange resin, $\varphi$4.6 mm × 60 mm (Hitachi High-Tech Science Corp., Tokyo, Japan); the mobile phase was composed of protein hydrolysate analysis buffer (PH Kanto, PH-1, PH-2, PH-3, PH-4, and PJ-RG, Kanto Chemical Co., Inc., Tokyo, Japan); a Ninhydrin Coloring Solution kit for HITACHI (Fujifilm Wako Pure Chemical Corp., Osaka, Japan) was used as the coloring kit for analysis; flow rate was 0.40 mL/min; detection wavelengths were 570 nm (for detection of all amino acids except proline) and 440 nm (for the detection of proline).

To quantify cystine and methionine, 0.5 g of walnut kernels was mixed with 10 mL of formic acid and subjected to oxidation treatment for 16 h at 4 °C in a refrigerator. Subsequently, the mixture was vacuum concentrated and dried. Then, 50 mL of 20% hydrochloric acid was added, and hydrolysis was performed at temperatures ranging from 130 °C to 140 °C for 20 h. Following this, the protein hydrolysate was brought up to 100 mL with deionized water, and a 2 mL aliquot was removed for vacuum concentration and drying. Next, 20 mL of sodium citrate buffer (pH of 2.2) was added to the dried sample to prepare the test solution, which was analyzed using an LA8080 high-speed amino acid analyzer. The analytical conditions for the high-speed amino acid analyzer were set as follows: the column consisted of Hitachi custom ion-exchange resin, $\varphi$4.6 mm × 60 mm; the mobile phase was protein hydrolysate analysis buffer (PH Kanto, PH-1); the Hitachi ninhydrin color reagent kit was used as the measuring reagent; a flow rate of 0.35 mL/min was maintained, and detection wavelength was 570 nm.

For the quantification of tryptophan, 0.2 g of walnut kernels was combined with 7.8 g of barium hydroxide, 4.5 mL of water, and 0.5 mL of 60% thiodiethylene glycol. The mixture was heated at 110 °C for 12 h to facilitate dissolution. Following this, 20% hydrochloric acid was added for neutralization. Subsequently, a 3 mol/L sodium hydroxide solution was added to bring the volume to 100 mL to prepare the test solution. Prepared solutions were analyzed using high-performance liquid chromatography coupled with fluorescence detection (HPLC-FL, Shimadzu Corp., Kyoto, Japan). Analytical conditions were configured as follows: the column was a Capcell Pak c18 AQ, $\varphi$4.6 mm × 250 mm (Osaka Soda Co., Ltd., Osaka, Japan); the mobile phase consisted of a mixed solution of 20 mol/L hypochlorous acid and methanol in an 8: 2 ratio; the flow rate was 0.7 mL/min; the fluorescence excitation wavelength was 285 nm; the fluorescence measurement wavelength was 348 nm; and the column temperature was 40 °C.

*2.5. Measurement of the Total Polyphenol Content*

To determine the total polyphenol content, 5 g of walnut kernels was first subjected to defatting with hexane according to the method described by Pico et al. [28] and the Folin–Ciocalteu method as described by Larrauri et al. [29] were used to quantify total polyphenol content. Gallic acid was used as standard for quantification.

*2.6. Measurement of the Mineral Content*

To quantify the minerals, including sodium (Na), magnesium (Mg), potassium (K), calcium (Ca), chromium (Cr), manganese (Mn), iron (Fe), copper (Cu), zinc (Zn), and selenium (Se), a slight modification of the method described by Azevedo et al. and Cindric et al. was used [30,31]. Initially, 10 mg of walnut kernels was placed in a polytetrafluoroethylene decomposition vessel. To this vessel, 6 mL of nitric acid and 1 mL of hydrogen peroxide were added. The mixture was subjected to microwave digestion using a TOPWave (Analytik Jena GmbH, Jena, Germany). The digestion process began at a temperature of 50 °C and gradually reached a maximum temperature of 200 °C after 30 min. Following digestion, the sample was diluted to 20 mL with deionized water. The mineral content was quantified using inductively coupled plasma mass spectrometry (ICP-MS, Agilent

8800 Triple Quadrupole ICP-MS, Santa Clara, CA, USA). The analytical conditions of ICP-MS were configured as follows: a nebulizer gas flow of 1.1 L/min, auxiliary gas flow of 0.9 L/min, plasma gas flow of 15.0 L/min, ICP RF (inductively coupled plasma radio frequency) power of 1550 W, analog HV (analog high-voltage) of 2535 V, pulse HV (pulse high-voltage) of 1734 V, cell entrance voltage of −40 V, and cell exit voltage of −70 V. Each detected mineral was quantified using a standard solution (ICP Standard Solution H, Kanto Chemical Co., Inc., Tokyo, Japan) and an internal standard (ICP Indium Standard Solution, Kanto Chemical Co., Inc., Tokyo, Japan).

### 2.7. Statistical Analysis

Group statistical comparisons were conducted using one-way analysis of variance (ANOVA) in conjunction with Tukey's multiple comparison test as appropriate. Statistical significance was set at $p < 0.05$. All statistical analyses were conducted using EZR (version 1.61), a software package developed at Saitama Medical University (Iruma-gun, Saitama, Japan) [32].

### 3. Results and Discussion

In this study, we examined the total lipid content and fatty acid composition of the kernels of four walnut species: English walnut, Shinano walnut, Oni walnut, and Hime walnut (Table 2 and Figure 2A). The total lipid contents were higher in the order of English walnut, Shinano walnut, Oni walnut, and Hime walnut, respectively (Table 2).

**Table 2.** Fatty acid composition of walnut kernels.

| Constituent | Walnuts | | | | | | | |
|---|---|---|---|---|---|---|---|---|
| | **English Walnut** | | **Shinano Walnut** | | **Oni Walnut** | | **Hime Walnut** | |
| | **Mean** | **SD** | **Mean** | **SD** | **Mean** | **SD** | **Mean** | **SD** |
| *Fatty acids (%)* | | | | | | | | |
| C14:0 | 0.05 | 0.00 | ND | | ND | | ND | |
| C16:0 | 5.08 [a] | 0.01 | 5.06 [a] | 0.01 | 2.33 [b] | 0.12 | 2.09 [c] | 0.02 |
| C18:0 | 2.22 [a] | 0.02 | 2.84 [b] | 0.01 | 0.61 [c] | 0.02 | 0.69 [c] | 0.01 |
| C20:0 | 0.08 | 0.00 | 0.09 | 0.00 | ND | | ND | |
| Total SFA | 7.42 | 0.02 | 8.10 | 0.00 | 2.99 | 0.15 | 2.84 | 0.04 |
| C16:1 ω7 | 0.06 | 0.02 | ND | | 0.05 | 0.03 | ND | |
| C18:1 ω9 | 12.29 [a] | 0.06 | 21.76 [b] | 0.05 | 8.66 [c] | 0.32 | 14.58 [d] | 0.06 |
| C20:1 ω9 | 0.25 [a] | 0.06 | 0.17 [b] | 0.01 | 0.19 [a] | 0.02 | 0.22 [a] | 0.00 |
| C22:1 ω9 | ND | | 0.11 | 0.12 | 0.22 | 0.02 | ND | |
| Total MUFA | 13.18 | 0.71 | 22.09 | 0.12 | 9.13 | 0.28 | 14.89 | 0.05 |
| C18:2 ω6 | 62.66 [a] | 0.58 | 61.95 [a] | 0.17 | 74.21 [b] | 0.32 | 73.17 [c] | 0.05 |
| C18:3 ω3 | 16.58 [a] | 0.16 | 7.81 [b] | 0.05 | 13.60 [c] | 0.11 | 9.05 [d] | 0.05 |
| Total PUFA | 79.24 | 0.71 | 69.76 | 0.15 | 87.81 | 0.40 | 82.21 | 0.01 |
| *Ratios* | | | | | | | | |
| SFA: MUFA: PUFA | 7.42: 13.18: 79.24 | | 8.04: 22.22: 69.56 | | 2.99: 9.13: 87.81 | | 2.84: 14.89: 82.21 | |
| ω6/ω3 | 3.78 | | 7.93 | | 5.45 | | 8.09 | |
| MUFA/SFA | 1.78 | | 2.75 | | 3.05 | | 5.25 | |
| UFA/SFA | 12.45 | | 11.43 | | 32.42 | | 34.24 | |

Values are mean ± SD ($n = 4$). Averages with equal letters in the same column do not differ at the 5% level of significance for the Tukey's multiple comparison test ($p > 0.05$). For example, there are no significant differences between *a* and *a*, *b* and *b*, *c* and *c*, *d* and *d*, meaning that groups with non-overlapping letters are significantly different. Components that did not show significant differences between groups have no letters. No trans fatty acids were detected in any walnut species. MUFA, monounsaturated fatty acids; ND, not detected; PUFA, polyunsaturated fatty acids; SFA, saturated fatty acids; UFA, unsaturated fatty acids.

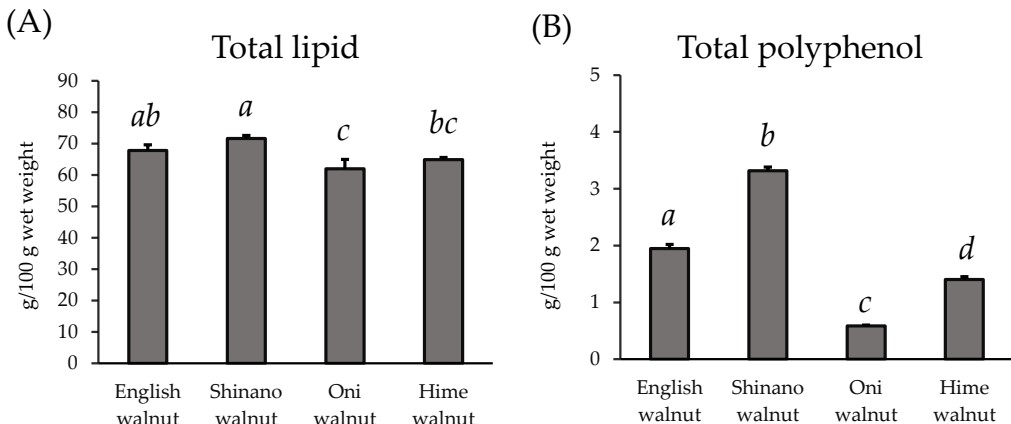

**Figure 2.** (**A**) Total lipid contents and (**B**) total polyphenol contents of walnut kernels. All values are presented in g/100 g wet weight. The values are mean ± SD (*n* = 3–4). Averages with equal letters in the same column do not differ at the 5% level of significance for the Tukey's multiple comparison test (*p* > 0.05). For example, there are no significant differences between *a* and *a*, *b* and *b*, *c* and *c*, *d* and *d*, meaning that groups with non-overlapping letters are significantly different.

The primary fatty acid found in walnuts is linoleic acid (C18:2 ω6), which had an approximately 10% higher composition in Oni walnut and Hime walnut compared to English walnut and Shinano walnut. In contrast, palmitic acid (C16:0) and stearic acid (C18:0) tended to be lower in Oni walnut and Hime walnut, although the difference was not as pronounced as that for linoleic acid. Significant differences were found in α-linolenic acid (C18:3 ω3) among all walnut species, with the concentrations from highest to lowest as follows: English walnut, Oni walnut, Hime walnut, and Shinano walnut. The order from highest to lowest for oleic acid (C18:1 ω9) was Shinano walnut, Hime walnut, English walnut, and Oni walnut. The ratio of daily fatty acid intake, particularly the ω6/ω3 ratio, has significant implications for human health [33]. The recommended optimal ratio for a healthy diet falls within the range of 1 to 4 for ω6/ω3, with an emphasis on increasing ω3 intake while reducing the intake of ω6 fatty acids to achieve this balance [34]. However, modern Western diets exhibit a ω6/ω3 ratio exceeding 20 [35]. In our study, the ω6/ω3 ratios for the examined walnut species were relatively low for English walnut, Shinano walnut, Oni walnut, and Hime walnut, respectively. These results suggest that walnuts could serve as valuable dietary source for supplementing ω3 fatty acids and adjusting the ω6/ω3 ratio. Along this line, other researchers have been exploring the incorporation of walnut paste into fermented foods and meats to effectively increase the ω3 fatty acid composition [36,37].

Unsaturated fatty acids (UFA), which are abundant in the Mediterranean diet, are believed to reduce the risk of obesity and cardiovascular disease. Therefore, replacing saturated fatty acids (SFA) with UFA to improve the quality of dietary fats is important [38]. In our study, Oni walnut and Hime walnut were approximately threefold higher in these fats than Shinano walnut and English walnut. Furthermore, (monounsaturated fatty acid (MUFA)/SFA) ratios were 1.78, 2.75, 3.05, and 5.25, for English walnut, Shinano walnut, Oni walnut, and Hime walnut, respectively. These findings highlight that the Hime walnut, a native species, contains nearly three times the MUFA of the English walnut. As such, the data suggest that Oni walnut and Hime walnut could serve as rich sources of UFA.

In our study, we measured the total protein content in kernels of four walnut species: English walnut, Shinano walnut, Oni walnut, and Hime walnut, and the amino acid content in the kernels of three species: Shinano walnut, Oni walnut, and Hime walnut (Table 3). An intriguing observation emerged when comparing Japanese native walnuts species, Oni walnut (25.23 g/100 g wet weight) and Hime walnut (24.45 g/100 g wet weight), to cultivated species, English walnut (13.78 g/100 g wet weight) and Shinano walnut (15.88 g/100 g wet weight). The native species stood out, with approximately 10 g/100 g

wet weight higher total protein content. This trend is consistent with the high total protein content of 24.10 g/100 g wet weight of the Black walnut, a native species in the United States [39]. These findings suggest that native walnut species generally contain more protein than cultivars. When examining individual amino acids, Oni walnut and Hime walnut exhibited richer profiles than English walnut and Shinano walnut. Among the essential amino acids, isoleucine had an approximately 0.40 g higher content in the native species compared to the cultivars. Additionally, histidine, leucine, lysine, methionine, phenylalanine, threonine, tryptophan, and valine exhibited 0.20–0.50 g higher contents in the native species compared to the cultivars. Among the non-essential amino acids, glutamic acid and arginine were particularly prominent, with 1.50–2.00 g higher contents in the native species compared to the cultivars. Other non-essential amino acids, such as alanine, aspartic acid, cystine, glycine, proline, serine, and tyrosine, were also present in with 0.20–1.10 g higher contents in the native species compared to the cultivars.

These results emphasize that Japanese native species, Oni walnut and Hime walnut, are rich sources of proteins and amino acids. Essential amino acids such as isoleucine, leucine, and valine, which are relatively abundant in these walnut species, play crucial roles in human protein synthesis [40]. Furthermore, non-essential amino acids such as glutamic acid and arginine are known to contribute to the improvement of neurodegenerative and inflammatory diseases [41]. Consequently, these walnut species have significant potential in the functional food and dietary supplement industries. Native walnut species may address the increasing global demand for plant-based proteins. Conversely, according to the amino acid scoring patterns established by the World Health Organization (WHO)/Food and Agriculture Organization of the United Nations (FAO)/United Nations University (UNU) expert consultation, these walnut species tend to have lower lysine content [42]. Lysine deficiency may be compensated for by incorporating other lysine-rich food sources, such as legumes, grains and seeds, into the diet [43].

We are also in the process of determination of carbohydrates in English walnut, Shinano walnut, Oni walnut, and Hime walnut (Supplemental Information: Carbohydrate contents of walnut kernels). Sucrose is known as one of the major carbohydrates in English walnut [45]. Although still in preliminary trials, we are finding that sucrose content is higher in English walnut (1.52 g/100 g wet wt), Shinano walnut (1.29 g/100 g wet wt), Oni walnut (1.06 g/100 g wet wt), and Hime walnut (0.99 g/100 g wet wt) in that order (Supplemental Information). These results suggest that the carbohydrate content of cultivated walnuts, Oni walnut, and Hime walnut may differ, but more detailed analysis is essential in the future.

Furthermore, we analyzed the mineral content of the kernels of English walnut, Shinano walnut, Oni walnut, and Hime walnut (Table 4). Walnuts are a rich source of magnesium, and our analysis confirmed that magnesium was the most abundant mineral in all the walnut species. Specifically, Oni walnut (315.59 mg/100 g wet weight) and Hime walnut (319.10 mg/100 g wet weight) contained approximately twice as much magnesium as English walnut (154.03 mg/100 g wet weight). Magnesium plays vital role in activation and maintenance of enzymes in the human body. Numerous studies have reported associations between magnesium deficiency and various health issues, including diabetes, ischemic heart disease, hypertension, periodontal disease, chronic fatigue, constipation, cancer, systemic inflammatory diseases, and dementia [46], and magnesium deficiency is a global concern [47]. A recent survey study conducted by Cindric et al. indicated that approximately 43% of the United States population did not have the recommended magnesium intake levels [31].

**Table 3.** Total protein and amino acid contents of walnut kernels with the measurements collected twice.

| Constituent | Walnuts | | | |
|---|---|---|---|---|
| | **English Walnut** | **Shinano Walnut** | **Oni Walnut** | **Hime Walnut** |
| | **Mean (Measured Value)** | **Mean (Measured Value)** | **Mean (Measured Value)** | **Mean (Measured Value)** |
| *Total protein* (g/100 g wet wt) | 13.78 (13.75, 13.80) | 15.88 (15.88, 15.88) | 25.23 (24.98, 25.59) | 24.45 (24.51, 24.39) |
| *Amino acids* (g/100 g wet wt) | | | | |
| Alanine (Ala) | 0.67 † | 0.68 (0.69, 0.67) | 1.14 (1.14, 1.14) | 1.11 (1.11, 1.12) |
| Arginine (Arg) | 2.28 † | 2.40 (2.40, 2.39) | 4.06 (4.06, 4.06) | 3.89 (3.87, 3.92) |
| Aspartic acid (Asp) | 1.83 † | 1.58 (1.59, 1.57) | 2.70 (2.68, 2.71) | 2.62 (2.59, 2.64) |
| Cystine (Cyss) | 0.21 † | 0.27 (0.27, 0.27) | 0.40 (0.40, 0.40) | 0.38 (0.38, 0.38) |
| Glutamic acid (Glu) | 2.82 † | 3.31 (3.37, 3.24) | 5.37 (5.32, 5.42) | 5.17 (5.15, 5.19) |
| Glycine (Gly) | 0.82 † | 0.84 (0.85, 0.83) | 1.31 (1.31, 1.31) | 1.20 (1.20, 1.21) |
| Histidine (His) | 0.39 † | 0.39 (0.39, 0.39) | 0.70 (0.70, 0.70) | 0.67 (0.67. 0.68) |
| Isoleucine (Ile) | 0.63 † | 0.60 (0.61, 0.59) | 0.99 (0.99, 0.99) | 0.96 (0.96, 0.97) |
| Leucine (Leu) | 1.17 † | 1.13 (1.14, 1.12) | 1.84 (1.84, 1.84) | 1.79 (1.77, 1.80) |
| Lysine (Lys) | 0.42 † | 0.43 (0.44, 0.42) | 0.65 (0.65, 0.65) | 0.65 (0.65, 0.64) |
| Methionine (Met) | 0.24 † | 0.25 (0.24, 0.25) | 0.52 (0.52, 0.52) | 0.50 (0.50, 0.50) |
| Phenylalanine (Phe) | 0.71 † | 0.70 (0.70, 0.69) | 1.22 (1.22, 1.22) | 1.18 (1.17, 1.19) |
| Proline (Pro) | 0.71 † | 0.55 (0.55, 0.55) | 0.96 (0.96, 0.96) | 0.94 (0.94, 0.95) |
| Serine (Ser) | 0.93 † | 0.85 (0.86, 0.85) | 1.38 (1.38, 1.38) | 1.32 (1.32, 1.33) |
| Threonine (Thr) | 0.60 † | 0.53 (0.54, 0.53) | 0.84 (0.84, 0.85) | 0.82 (0.82, 0.83) |
| Tyrosine (Tyr) | 0.41 † | 0.52 (0.52, 0.51) | 0.85 (0.84, 0.85) | 0.81 (0.80, 0,82) |
| Tryptophan (Trp) | 0.17 † | 0.20 (0.20, 0.19) | 0.36 (0.36, 0.36) | 0.40 (0.41, 0.40) |
| Valine (Val) | 0.75 † | 0.72 (0.73, 0.71) | 1.25 (1.26, 1.24) | 1.22 (1.22, 1.23) |

† means that the data from reference [44].

Potassium content was approximately 1.5 times higher in the Japanese native species, Oni walnut (664.04 mg/100 g wet weight) and Hime walnut (675.96 mg/100 g wet weight), than in the cultivated species, English walnut (413.57 mg/100 g wet weight) and Shinano walnut (393.27 mg/100 g wet weight). In plants, potassium is involved in the activation of proton pumps, which facilitate the accumulation of sucrose and sorbitol in roots and seeds [48,49]. The distinctive taste of Oni walnut and Hime walnut may be influenced by their high potassium concentrations [14–16].

**Table 4.** Mineral content of walnut kernels.

| Constituent | Walnuts | | | | | | | |
| --- | --- | --- | --- | --- | --- | --- | --- | --- |
| | English Walnut | | Shinano Walnut | | Oni Walnut | | Hime Walnut | |
| | Mean | SD | Mean | SD | Mean | SD | Mean | SD |
| *Minerals (mg/100 g wet wt)* | | | | | | | | |
| Sodium (Na) | 3.85 | 2.13 | 2.52 | 0.40 | 3.79 | 0.89 | 2.82 | 0.66 |
| Magnesium (Mg) | 154.03 [a] | 2.68 | 164.59 [a] | 2.43 | 315.59 [b] | 12.69 | 319.10 [b] | 1.86 |
| Potassium (K) | 413.57 [a] | 4.39 | 393.27 [a] | 9.52 | 664.04 [b] | 100.54 | 675.96 [b] | 2.82 |
| Calcium (Ca) | 118.53 [a] | 24.39 | 64.51 [b] | 1.22 | 83.08 [c] | 10.20 | 88.74 [c] | 0.68 |
| Chromium (Cr) | 0.16 [ab] | 0.06 | 0.66 [c] | 0.09 | 0.35 [bd] | 0.11 | 0.20 [ad] | 0.09 |
| Manganese (Mn) | 2.34 [a] | 0.04 | 1.50 [b] | 0.01 | 7.18 [c] | 0.27 | 7.14 [c] | 0.07 |
| Iron (Fe) | 3.85 [a] | 0.48 | 8.49 [b] | 0.86 | 8.26 [bc] | 2.35 | 6.41 [abc] | 0.72 |
| Copper (Cu) | 1.51 [a] | 0.03 | 1.60 [b] | 0.03 | 1.66 [b] | 0.03 | 1.83 [c] | 0.04 |
| Zinc (Zn) | 2.21 [a] | 0.28 | 1.32 [b] | 0.02 | 2.41 [c] | 0.13 | 2.66 [c] | 0.07 |
| Selenium (Se) | 0.01 [ab] | 0.00 | 0.01 [b] | 0.00 | 0.002 [a] | 0.00 | 0.002 [a] | 0.00 |

Values are mean $\pm$ SD ($n = 3$). Averages with equal letters in the same column do not differ at the 5% level of significance for the Tukey's multiple comparison test ($p > 0.05$). For example, there are no significant differences between *a* and *a*, *b* and *b*, *c* and *c*, *d* and *d*, meaning that groups with non-overlapping letters are significantly different. Components that did not show significant differences between groups have no letters.

Manganese contents in Oni walnut (7.18 mg/100 g wet weight) and Hime walnut (7.14 mg/100 g wet weight) were approximately three times higher than that of the cultivated species, English walnut (2.34 mg/100 g wet weight) and Shinano walnut (1.50 mg/100 g wet weight). When soil is deficient in manganese, lipid synthesis in the plant decreases [50]. Although this study confirmed higher manganese contents in Japanese native walnut species, this may be due to differences in soluble manganese concentrations in the soil. Walnut species rich in manganese may accumulate higher concentrations of α-linolenic acid (C18:3 ω3) in their leaves, which in turn could increase the production of jasmonic acid, a plant hormone-like component [51]. This may enhance plant resistance to pests and diseases. The fatty acid composition variation among walnut species, especially the high proportion of α-linolenic acid (C18:3 ω3) in the kernels of Oni walnut and Hime walnut, might also be influenced by this pathway. Future research should explore the relationship between the fatty acid composition of walnut species and the soil manganese concentrations.

Iron content of Oni walnut (8.26 mg/100 g wet weight) and Hime walnut (6.41 mg/100 g wet weight) was approximately twice that of the English walnut (3.85 mg). Iron is an essential mineral in the human body that is crucial for energy production and oxygen transport. Iron can potentially accelerate the oxidation of coexisting oils during cooking [52,53]. However, by optimizing conditions such as storage method, cooking technique, types of coexisting oils, and coexisting antioxidants, this effect can be minimized, thereby maximizing the benefits of walnuts as an iron-rich food source.

Total polyphenol contents in the kernels of the four walnut species are shown in Figure 2B. Daily polyphenol intake is believed to have potential health benefits [54,55]. Significant differences in total polyphenol content were observed among walnut species. In particular, Oni walnut (585.92 mg/100 g wet weight) and Hime walnut (1403.49 mg/100 g wet weight) exhibited significantly lower polyphenol contents compared to English walnut

(1946.99 mg/100 g wet weight) and Shinano walnut (3318.33 mg/100 g wet weight). This suggests that Oni walnut and Hime walnut have relatively low total polyphenol contents owing to their reduced astringency from phenolic components [14,16]. As indicated in Table 1 and Figure 1, the color of Oni walnut kernels in this study was "light amber", that of Hime walnut was "light", while the other walnuts were "extra light". Several studies have found a negative correlation between the darkness of walnut kernel color and total polyphenol content [56,57], supporting our findings that Oni walnut, with the darkest color, had the lowest total polyphenol content. Although the exact mechanism remains unclear, oxidation reactions of polyphenolic compounds frequently occur in high moisture walnuts, which may result in the formation of colored oxidation products that cause the kernels to turn brown [58]. Additionally, the polyphenol content of walnut kernels may vary depending on agricultural practices and post-harvest processing methods [59]. Therefore, not only species-specific characteristics, but also environmental factors, may significantly influence the polyphenol content. Further detailed investigations are required to understand these factors.

The results of our study summarizing the general composition of the kernels of the four walnut species (English walnut, Shinano walnut, Oni walnut, and Hime walnut) are shown in Figure 3. One significant factor that may account for these differences is the growth climate. According to the report from FAO, Japan experiences an annual precipitation of 1688 mm, approximately 2.5 times higher than that of the United States (715 mm) [60,61]. Additionally, average annual temperature in Yamagata prefecture in Japan, stands at 11.7 °C, notably lower than that in California in the United States (17.5 °C) [61,62]. The average annual sunshine duration in Japan is 1613 h, which is considerably lower than that of California (3608 h), and the relative humidity in Japan is 74%, in contrast to California's 62% [63–65]. It is hard to consider the mechanisms in any convincing detail, e.g., the effect of harvesting location, climatic conditions, cultivation or other indicators, on the basis of the results of the present study alone. Furthermore, Amaral et al. reported that the vitamin E content varied from year to year, even for walnuts grown under the same agricultural practices in the same experimental fields for three years [66]. In this study, only walnut species harvested in 2022 were measured, but it would be essential to obtain experimental data over time as a future task for this study. These distinct climatic conditions in Japan may subject walnut species to increased exogenous stress, potentially influencing the composition of Japanese native walnut species. Further comprehensive investigations in the future are warranted to elucidate this relationship. Recently, there has been a reevaluation of the nutritional value of native walnuts, including Black walnut. This study contributes significantly to enhancing the nutritional value of walnuts by furnishing a comprehensive profile of the constituents of Japanese native walnut species.

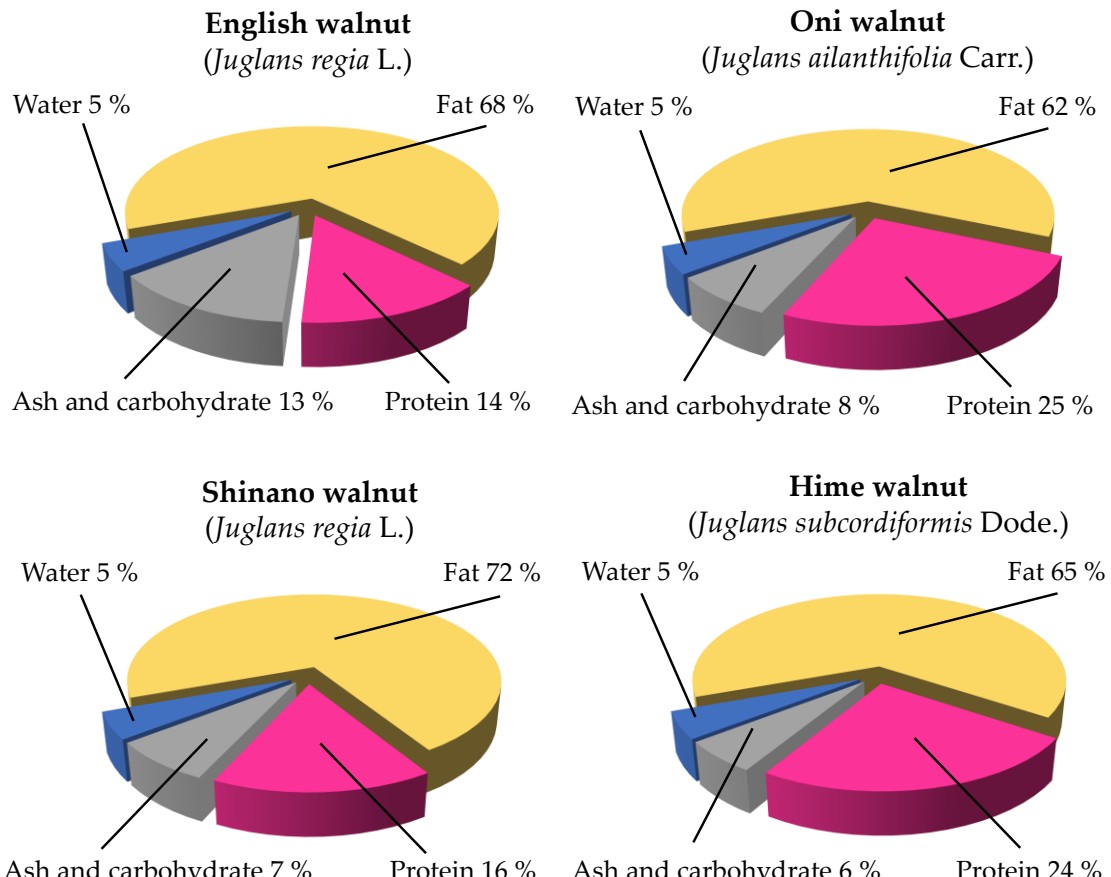

**Figure 3.** General composition of the kernels of the four walnut species measured in present study (English, Shinano, Oni, and Hime walnuts). These data were calculated from the results of this study.

## 4. Conclusions

In conclusion, this study aimed to assess the potential utility of Japanese native walnut species by quantifying and comparing various nutritional aspects, including total lipid content, fatty acid composition, total protein content, amino acid content, total polyphenol content, and mineral content among four walnut species (English walnut, Shinano walnut, Oni walnut, and Hime walnut). Notably, Japanese native species, Oni walnut and Hime walnut, exhibited significant disparities in their constituents when juxtaposed with cultivars. These native walnuts have a lower fat content and higher protein and essential mineral content, accompanied by a high content of unsaturated fatty acids (UFA). Overall, this study offers new insights into the components of Oni walnut and Hime walnut, shedding light on their nutritional value and potential applications. These native walnut species hold promise as valuable dietary resources and warrant further exploration in the context of nutrition and food science. This study will contribute to the development of functional and processed foods by encouraging increased production of native walnut species, which are rich in protein, unsaturated fatty acids and minerals, and their use in local cuisines and health-promoting foods.

**Supplementary Materials:** The following supporting information can be downloaded at: https://www.mdpi.com/article/10.3390/horticulturae9111221/s1, Table S1: Carbohydrate contents of walnut kernels. Reference [67] is cited in the supplementary materials.

**Author Contributions:** R.F., T.M. and M.T. conceived the study. R.F., C.A. and M.B. performed experiments. R.F., T.M., C.A., M.B. and M.T. wrote the paper. T.M. and M.T. supervised the study. All authors have read and agreed to the published version of the manuscript.

**Funding:** This work was supported by the Tohoku University Fund.

**Institutional Review Board Statement:** Not applicable.

**Informed Consent Statement:** Not applicable.

**Data Availability Statement:** The data that support the findings of this study are available from the corresponding authors, upon reasonable request.

**Acknowledgments:** We thank Atsushi Kawauchi (International Research Institute of Disaster Science, Tohoku University) for his support in the use of the freeze dryer. We thank Shinji Takahashi (Technical Division, School of Engineering, Tohoku University) for his support in the ICP-MS analysis. We thank Ohki Higuchi (New Industry Creation Hatchery Center (NICHe), Tohoku University) for his support in the HPLC-ELSD analysis.

**Conflicts of Interest:** The authors declare no conflict of interest.

## Abbreviations

Analog HV, analog high-voltage; ANOVA, analysis of variance; Ala, alanine; Arg, arginine; Asp, aspartic acid; C14:1, myristic acid; C16:1 ω7, palmitoleic acid; C16:1, palmitic acid; C18:0, stearic acid; C18:1 ω9, oleic acid; C18:2 ω6, linoleic acid; C18:3 ω3, α-linolenic acid; C20:0, arachidonic acid; C20:1 ω9, gondoic acid; C22:1 ω9, erucic acid; Ca, calcium; Cr, chromium; Cu, copper; Cys, cysteine; EDTA, ethylenediaminetetraacetic acid; Fe, iron; GC-FID, gas chromatograph-flame ionization detector; Glu, glutamic acid; Gly, glycine; HPLC, high-performance liquid chromatography; HPLC-FL, high-performance liquid chromatography-fluorescence detector; His, histidine; ICP RF, inductively coupled plasma radio frequency; ICP-MS, inductively coupled plasma mass spectrometry; ICP-OES, inductively coupled plasma optical emission spectroscopy; Ile, isoleucine; K, potassium; Leu, leucine; Lys, lysine; MUFA, monounsaturated fatty acids; Met, methionine; Mg, magnesium; Mn manganese; Na, sodium; NaCl, sodium chloride; PUFA, polyunsaturated fatty acids; Phe, phenylalanine; Pro, proline; Pulse HV, pulse high-voltage; SFA, saturated fatty acids; Se, selenium; Ser, serine; Thr, threonine; Trp, tryptophan; Tyr, tyrosine; UFA, unsaturated fatty acids; Val, valine; Zn, zinc; ω3, ω3 fatty acids; ω6, ω6 fatty acids; ω9, ω9 fatty acids.

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
