# Peer review of "Quantification and Comparison of Nutritional Components in Oni Walnut (Juglans ailanthifolia Carr.), Hime Walnut (Juglans subcordiformis Dode.), and Cultivars"

_horticulturae, doi:10.3390/horticulturae9111221_

Round 1

Reviewer 1 Report

Comments and Suggestions for Authors

Line 127: separate 1 uL

Line 136: Nowhere else in the manuscript does the meaning of AOAC appear

In some lines it unites and in others it separates numerical value and units, please review and modify homogeneously

In Figure 2, the part that specifies the sample size should be in the figure description and not in the figure.

Author Response

For the Reviewer 1

 We appreciate your valuable time and effort in reviewing this article. We found that all of your comments are necessary for our article. We send a revised manuscript with track changes in response to your comments. We would be grateful if you could confirm these.

Comment 1 from Reviewer 1

Line 127: separate 1 μL

Answer for Comment 1 from Reviewer 1

Thank you for pointing this out. We have changed line 174 in the revised manuscript to "1 μL".

Comment 2 from Reviewer 1

Line 136: Nowhere else in the manuscript does the meaning of AOAC appear.

Answer for Comment 2 from Reviewer 1

Thank you for your comment. We used the term "AOAC" in the experimental section "2.3. Measurement of total protein content" but this was absurd. Therefore, we have removed the unneeded text in line 183-184 of the revised manuscript.

Comment 3 from Reviewer 1

In some lines it unites and in others it separates numerical value and units, please review and modify homogeneously.

Answer for Comment 3 from Reviewer 1

Thank you for your comment.

As reviewer 1 commented, there was a mix of lines in the text where numbers and units were integrated and separated. Therefore, we have again reviewed the whole article and unified the numbers and units to separate them.

Comment 4 from Reviewer 1

In Figure 2, the part that specifies the sample size should be in the figure description and not in the figure.

Answer for Comment 4 from Reviewer 1

Thanks for pointing this out. We agree with reviewer 1's comment. We have removed the sample size information in the figure 2 and added it in the figure caption in line 315 of the revised manuscript.

Reviewer 2 Report

Comments and Suggestions for Authors

This manuscript is of rather low value and is just a simple report showing the composition of nuts harvested in 2022 in Japan and partly in California. The paper lacks a planned experiment, an analysis of the impact of the harvesting site, climatic conditions, cultivation or other indicators. Collecting 4 types of nuts in one year and analysing their composition is of no scientific value and does not solve any scientific problem.

The paper is well written and showing the results from, for example, successive harvest years would have enriched it considerably. However, showing the same results in graph and table form is unnecessary (Fig. 2A and Tab. 2). I believe that the paper should be withdrawn as it could be preliminary research for a larger project and in this form has very little scientific value.

Comments on the Quality of English Language

English language is correct.

Author Response

For the Reviewer 2

Thank you very much for reviewing this article. All the comments from reviewer 2 are very important and essential for this article. Through your comment, authors have learned a lot about the lack of this research. Thank you again for spending your valuable time on this review. We revised this article according to Reviewer 2’s comments with track changes. We would like you to confirm the revised manuscript.

Comment 1 from Reviewer 2

This manuscript is of rather low value and is just a simple report showing the composition of nuts harvested in 2022 in Japan and partly in California. The paper lacks a planned experiment, an analysis of the impact of the harvesting site, climatic conditions, cultivation or other indicators. Collecting 4 types of nuts in one year and analysing their composition is of no scientific value and does not solve any scientific problem.

Answer for Comment 1 from Reviewer 2

Thank you for pointing this out. As reviewer 2 said, we thought this article insufficiently detailed the impact of the planned experiments, harvest location, climatic conditions, cultivation, and other indexes. However, we believe the academic value of this study is high, as this is the first paper in the world to provide detailed nutritional composition of Oni walnut and Hime walnut, two native species that have been used in Japanese local cuisine for a long time. On the other hand, as reviewer 2 said, we also thought it was important to note the results of successive harvest years. This is an important research topic for the future, and we have added an explanation of its importance in lines 446-453 of the discussion, along with new references.

Comment 2 from Reviewer 2

The paper is well written and showing the results from, for example, successive harvest years would have enriched it considerably. However, showing the same results in graph and table form is unnecessary (Fig. 2A and Tab. 2). I believe that the paper should be withdrawn as it could be preliminary research for a larger project and in this form has very little scientific value.

Answer for Comment 2 from Reviewer 2

Thank you for pointing this out. As reviewer 2 said, there was duplication of data between figures and tables, which made it cumbersome. Therefore, the data for Total lipid in Table 2 were removed. This research is not a preliminary study for some larger project. This study was planned out of a desire to preserve the valuable food culture of the region. However, we would like to apply your valuable comments to our future research. Thank you also for your recognition that this paper is well written.

Reviewer 3 Report

Comments and Suggestions for Authors

Manuscript title: 

 Quantification and comparison of nutritional components in  Oni walnut (Juglans ailanthifolia Carr.), Hime walnut (Juglans subcordiformis Dode.), and cultivars  

This study has certain significance in walnut chemistry and especially a new cultivar ……... However, revisions are necessary for the current version of the manuscript. The following questions to be addressed/considered may be helpful to improve the manuscript.

Major comments

·       Insufficient Abstract: In the abstract, the main aim and background of the manuscript are missing, the current version it only highlights the result. In addition, it would be even better to have a sentence as a future perspective.

·       The unit/abbreviation is not mentioned before, consider defining the abbreviation when mentioned for the first time…. Please check throughout the manuscript to define the abbreviations.

·       Line 52-58, the aim or hypothesis of the study is clear, however, the approach is missing ….

·       Lake of scientific literature to support the statements and findings throughout the manuscript…... I have made some suggestions for that and more need it….

·       More information is needed for ALL TABLE captions and define the abbreviation and units that are used. And adjust the significant figures for the table and manuscript.

·       Grammar and punctuation issuers need to be addressed. I have selected/mentioned some as examples.

·       I am not sure whether the ‘’…..’’ term is well discussed in the abstract and manuscript. Please consider discussing it or rephrasing it.

·       I have a major concern about the results and discussion section. The authors describe the results and compare the results with previous studies, however, insight mechanisms are still insufficient.

·       The language is generally clear, with some exceptions where the authors are a bit too innovative with the terminology, although there are other good terms to use…..eg.

Specific comments:

Abstract

If the unit/abbreviation is not mentioned before, consider defining the abbreviation when mentioned for the first time.

Introduction:

Line 32-38: A complicated sentence, please revise and check the grammar

Line 44: A reference is needed here, for example, you can use:

https://doi.org/10.1007/s11295-017-1214-0

https://doi.org/10.1080/10942912.2015.1114951

Line 55: A reference is needed here, for example, you can use:

https://doi.org/10.1080/22297928.2016.1152912

Line 63: A reference is needed here….

Line 59-65: These are rather long sentences, better to break them down into more sentences.

In MM section

Literature references are missing for all sub-section. It would be better to cite the references that the procedure adopted.

Additional info is needed for the table caption, most importantly significant figures.

In MM section, what is the quality control (QC) data? There is no mention of the QC.

What is the accuracy of the instruments, recovery, LOD, and LOQ ……. These parameters are needed to report the efficiency of any analytical system.

In general, how many times you’ve recorded the data,? duplicate? Triplicate?..... what you mentioned in the text is not clear, please elaborate more on this

2.5. Statistical Analysis

How the comparison was made between the treatments? Ad see my comment for Figures

R&D section

These sections are repeating information already presented and explain things in an unnecessarily complicated way. The quality of the manuscript would benefit from the whole section being condensed, Line 241-258, Line 262-293, Line 343-369, Line 380-395…..

Figure 1: Please add a scale to the figure, maybe on the left side!

Figure 2. how the comparing made between the treatments and assigning the letter for the statistical difference is confusing. For example, Figure 1 a: How you can have ab, a; c, bc;? Please elaborate more, or consider changing the format.

Conclusion

I believe there are other important conclusions that could be made from this study…. And the future perspectives for the following research are highly crucial here.

Comments on the Quality of English Language

English should be improved as, at times, it is difficult to understand what the author wanted to express. Moreover, there are apparent grammatical mistakes. I provided some non-exhaustive examples in the comments, but the authors are advised to check the entire manuscript for readability and clarity of presentation.

Author Response

Comment from the Reviewer 3

Quantification and comparison of nutritional components in Oni walnut (Juglans ailanthifolia Carr.), Hime walnut (Juglans subcordiformis Dode.), and cultivars.

This study has certain significance in walnut chemistry and especially a new cultivar ……... However, revisions are necessary for the current version of the manuscript. The following questions to be addressed/considered may be helpful to improve the manuscript.

Answer to Reviewer 3

We highly appreciate Reviewer 3’s accurate comments for this article to be improvement of the quality. We made this article according to your comments with track changes. We would like to you confirm the revised manuscript.

Major comments

Comment 1 from Reviewer 3

Insufficient Abstract: In the abstract, the main aim and background of the manuscript are missing, the current version it only highlights the result. In addition, it would be even better to have a sentence as a future perspective.

Answer for Comment 1 from Reviewer 3

Thank you for pointing this out. As reviewer 3 commented, the abstract lacked the main aim, background and future perspective of the manuscript. Therefore, was revised in line 14-33 in the revised manuscript.

Comment 2 from Reviewer 3

The unit/abbreviation is not mentioned before, consider defining the abbreviation when mentioned for the first time…. Please check throughout the manuscript to define the abbreviations.

Answer for Comment 2 from Reviewer 3

Thank you for your comment. As you mentioned, this paper lacked an explanation of units and abbreviations. Therefore, we have created a "List of Abbreviations" and added it to lines 37-53 of the revised manuscript. In addition, we reviewed the whole article in detail and confirmed that there were no oversights in the explanation of the abbreviation definitions when it was first presented.

Comment 3 from Reviewer 3

Line 52-58, the aim or hypothesis of the study is clear, however, the approach is missing.

Answer for Comment 3 from Reviewer 3

Thank you for your comment. As reviewer 3 commented, this paragraph lacked information about the approach. Therefore, we add the information of approach in the lines 80-84 in the revised manuscript.

Comment 4 from Reviewer 3

Lake of scientific literature to support the statements and findings throughout the manuscript…... I have made some suggestions for that and more need it….

Answer for Comment 4 from Reviewer 3

Thank you for your comment. We have added the references you suggested. In addition, we have added new references that we found.

Comment 5 from Reviewer 3

More information is needed for ALL TABLE captions and define the abbreviation and units that are used. And adjust the significant figures for the table and manuscript.

Answer for Comment 5 from Reviewer 3

Thank you for your comment. We have created a "List of Abbreviations" and added it to lines 37-53 of the revised manuscript. In addition, the whole article was reviewed and the significant figures and abbreviations in the tables and manuscript were again adjusted and corrected.

Comment 6 from Reviewer 3

Grammar and punctuation issuers need to be addressed. I have selected/mentioned some as examples.

Answer for Comment 6 from Reviewer 3

Thank you for your comments regarding the grammar and punctuation issues. Reviewer 3's comments were very helpful.

Comment 7 from Reviewer 3

I am not sure whether the ‘’…..’’ term is well discussed in the abstract and manuscript. Please consider discussing it or rephrasing it.

Answer for Comment 7 from Reviewer 3

Thank you for your comments. As you pointed out, the article lacked information on the walnut species measured in this study. Therefore, we add more detailed information of Oni walnuts and Hime walnuts in line 90-99 of revised manuscript.

Comment 8 from Reviewer 3

I have a major concern about the results and discussion section. The authors describe the results and compare the results with previous studies, however, insight mechanisms are still insufficient.

Answer for Comment 8 from Reviewer 3

Thank you for your comments. As reviewer 3 said, it needs to describe more insight mechanisms in this article, such as harvest location, climatic conditions, cultivation, and other indexes. We had limited ability to present a considerably plausible explanation for the insight mechanism that could be considered based on the results of this study. However, we have added as much discussion as possible to lines 446-453 of the revised manuscript.

Comment 9 from Reviewer 3

The language is generally clear, with some exceptions where the authors are a bit too innovative with the terminology, although there are other good terms to use…..eg.

Answer for Comment 9 from Reviewer 3

Thanks for pointing this out. The entire paper has been reviewed by native speaker and we have corrected the overly innovative terminology to more general terms.

Specific comments

Comment 10 from Reviewer 3

If the unit/abbreviation is not mentioned before, consider defining the abbreviation when mentioned for the first time.

Answer for Comment 10 from Reviewer 3

Thank you for your comment. We reviewed the whole article in detail and confirmed that there were no oversights in the explanation of the abbreviation definitions when it was first presented.

Comment 11 from Reviewer 3

Line 32-38: A complicated sentence, please revise and check the grammar

Answer for Comment 11 from Reviewer 3

Thank you for pointing this out. We have corrected the grammar and made the text simpler.

Comment 12 from Reviewer 3

Line 44: A reference is needed here, for example, you can use:

https://doi.org/10.1007/s11295-017-1214-0

https://doi.org/10.1080/10942912.2015.1114951

Answer for Comment 12 from Reviewer 3

Thanks for your comment. We have added the reference you suggested (line 69 of the revised manuscript).

Comment 13 from Reviewer 3

Line 55: A reference is needed here, for example, you can use:

https://doi.org/10.1080/22297928.2016.1152912

Answer for Comment 13 from Reviewer 3

Thanks for your comment. We have added the reference you suggested (line 84 of the revised manuscript).

Comment 14 from Reviewer 3

Line 63: A reference is needed here….

Answer for Comment 14 from Reviewer 3

Thanks for your comment. We have added the new reference in line 101 of the revised manuscript.

Comment 15 from Reviewer 3

Line 59-65: These are rather long sentences, better to break them down into more sentences.

Answer for Comment 15 from Reviewer 3

Thank you very much for pointing this out. We checked the text again and divided this paragraph into two parts (line 90 to 112).

In MM section

Comment 16 from Reviewer 3

Literature references are missing for all sub-section. It would be better to cite the references that the procedure adopted.

Answer for Comment 16 from Reviewer 3

Thanks for your comment. We have added the new reference in each method of the revised manuscript.

Comment 17 from Reviewer 3

Additional info is needed for the table caption, most importantly significant figures.

Answer for Comment 17 from Reviewer 3

Thank you for pointing this out. We have added an explanation of the significant figures to each table caption.

Comment 18 from Reviewer 3

In MM section, what is the quality control (QC) data? There is no mention of the QC.

Answer for Comment 18 from Reviewer 3

Thanks for your comment. As reviewer 3 commented, a discussion on quality control of the samples used in this study is essential. Therefore, information on quality control has been added to line 124-126.

Comment 19 from Reviewer 3

What is the accuracy of the instruments, recovery, LOD, and LOQ ……. These parameters are needed to report the efficiency of any analytical system.

Answer for Comment 19 from Reviewer 3

Thank you for pointing this out. As reviewer 3 said, validation of analytical methods is important. In this experiment, we did not evaluate the accuracy of the equipment, recovery rate, LOD, or LOQ, but referred directly to already established analytical methods in the reference. Other quantitative studies of nutritional components of nuts already published in this journal do not seem to have considered these validations (Horticulturae 2023, 9, 1093. https://doi.org/10.3390/horticulturae9101093) (Horticulturae 2023, 9, 1101. https://doi.org/10.3390/horticulturae9101101) (Horticulturae 2023, 9, 1163. https://doi.org/10.3390/horticulturae9111163). In this study, the study was carried out in accordance with the parameters described in reference.

Comment 20 from Reviewer 3

In general, how many times you’ve recorded the data,? duplicate? Triplicate?..... what you mentioned in the text is not clear, please elaborate more on this

Answer for Comment 20 from Reviewer 3

Thank you for pointing this out. The data for Total Lipid was calculated from 5 measurements, with 3 measurements excluding the first and last. Total protein and amino acids were measured by duplicate. All other quantities were measured once. These descriptions were added to lines 126-129 of the revised manuscript.

2.5. Statistical Analysis

Comment 21 from Reviewer 3

How the comparison was made between the treatments? Ad see my comment for Figures

Answer for Comment 21 from Reviewer 3

Thank you for your comments. We have corrected the revisions to the figures and tables one by one based in your comments, as you can see in the comments below.

Comment 22 from Reviewer 3

These sections are repeating information already presented and explain things in an unnecessarily complicated way. The quality of the manuscript would benefit from the whole section being condensed, Line 241-258, Line 262-293, Line 343-369, Line 380-395…..

Answer for Comment 22 from Reviewer 3

Thanks for pointing this out. As reviewer 3 mentioned, these sentences were cumbersome and have been corrected to make the explanation more concise.

Comment 23 from Reviewer 3

Figure 1: Please add a scale to the figure, maybe on the left side!

Answer for Comment 23 from Reviewer 3

Thanks for your comment. The scale information is certainly important. We have replaced it with a new Figure 1 with the scale bar added, and would appreciate it if you could confirm.

Comment 24 from Reviewer 3

Figure 2. how the comparing made between the treatments and assigning the letter for the statistical difference is confusing. For example, Figure 1 a: How you can have ab, a; c, bc;? Please elaborate more, or consider changing the format.

Answer for Comment 24 from Reviewer 3

Thank you for pointing this out. Information regarding the letter for the statistical difference has been added to the captions of each figure and table in the revised manuscript.

Conclusion

Comment 25 from Reviewer 3

I believe there are other important conclusions that could be made from this study…. And the future perspectives for the following research are highly crucial here.

Answer for Comment 25 from Reviewer 3

Thank you for pointing this out. We have added a statement regarding the value of the findings of this study and a statement of future prospects to lines 481-484 of the revised manuscript.

Comment 26 from Reviewer 3

Comments on the Quality of English Language

English should be improved as, at times, it is difficult to understand what the author wanted to express. Moreover, there are apparent grammatical mistakes. I provided some non-exhaustive examples in the comments, but the authors are advised to check the entire manuscript for readability and clarity of presentation.

Answer for Comment 26 from Reviewer 3

Thanks for pointing this out. The whole article was reviewed by native speaker and we made the article more clarity of presentation.

Round 2

Reviewer 2 Report

Comments and Suggestions for Authors

Thank you for corrections. I suggest accept manuscript in present form.

Reviewer 3 Report

Comments and Suggestions for Authors

The revised manuscript has improved compared to the original version. The authors tried to address my questions as much as possible. I recommend the manuscript to be published!

Comments on the Quality of English Language

Overall, the quality of the written text with respect to English phrasing and grammar is good and acceptable.